# Surveillance of Healthcare-Associated Infections in the WHO African Region: Systematic Review of Literature from 2011 to 2024

**DOI:** 10.3390/antibiotics14121287

**Published:** 2025-12-18

**Authors:** Laetitia Gahimbare, Nathalie K. Guessennd, Claude Mambo Muvunyi, Walter Fuller, Sheick Oumar Coulibaly, Landry Cihambanya, Pierre Claver Kariyo, Olga Perovic, Ambele Judith Mwamelo, Diané Kouao Maxime, Valérie Gbonon, Konan Kouadio Fernique, Babacar Ndoye, Yahaya Ali Ahmed

**Affiliations:** 1World Health Organization, Regional Office for Africa, P.O. Box 06 Brazzaville, Congo; wfuller@who.int (W.F.); coulibalysh@who.int (S.O.C.); cihambanyal@who.int (L.C.); kariyop@who.int (P.C.K.); ndoyeb@who.int (B.N.); aliahmedy@who.int (Y.A.A.); 2Pasteur Institute of Côte d’Ivoire, 01 BP 490 Abidjan 01, Côte d’Ivoire; nathalieguessennd@pasteur.ci (N.K.G.); dianekouao@pasteur.ci (D.K.M.); valeriembengue@pasteur.ci (V.G.); ferniquekonan@pasteur.ci (K.K.F.); 3Rwanda Biomedical Centre, P.O. Box 7162 Kigali, Rwanda; claude.muvunyi@rbc.gov.rw; 4National Institute of Communicable Diseases (NICD), 1 Modderfontein Road, Johannesburg 2000, South Africa; olgap@nicd.ac.za; 5Green Templeton College, University of Oxford, UK. 50 Broad Street, Oxford Oxfordshire OX1, UK; abelejdith.mwamelo@gtc.ox.ac.uk

**Keywords:** healthcare–associated infection, hospital–acquired infection, nosocomial infection, infection control surveillance

## Abstract

Background: Evidence on HAIs in Africa is fairly common. Objectives: The main objective was to identify the surveillance tools used for healthcare–associated infections (HAIs) in countries in the WHO African Region. Secondary objectives focused on the organization of surveillance, the pathogens involved, and the frequency of multidrug–resistant species. Inclusion and exclusion criteria: Observational or interventional studies on healthcare–associated infections in humans, published between January 2011 and December 2024, in French or English, were included. However, the following publications were not included: animal studies, healthcare–associated infections not related to healthcare, literature reviews, studies outside the period or geographical area, and studies in languages other than French or English. Sources of information and search date: The databases consulted were PubMed, Web of Science, EMBASE, Cochrane, African Index Medicus, Google Scholar, and AJOL. The search was conducted between January and March 2025. Risk of bias assessment: The risk of bias was assessed using a specific grid (eleven criteria), scored from one (low) to three (high). The studies were classified into three levels of methodological quality. The results of the bias assessment showed that the publications were excellent (strong and moderate) with a cumulative rate of 99.9%. Methods of synthesizing results: Data were extracted using a standardized grid and synthesized narratively. No meta–analysis was performed. Number of studies and characteristics: 95 studies were included, mostly cross–sectional studies (82.1%), cohorts (10.4%), and a few case reports. Most were from West Africa (60.0%), particularly Nigeria (16.8%) and South Africa (14.7%). Main results: • Most common pathogens: *Staphylococcus aureus* (53.7%), *Escherichia coli* (43.2%), *Klebsiella pneumoniae* (32.6%). • Resistance profile: ESBL (27.4%), MRSA (21.1%), multidrug resistance (13.7%). • Sources of HAIs: mainly exogenous (83.2%). • Laboratory methods: phenotypic (70.5%), genotypic or genomic rare (3.1%). • Scope of studies: local (96.8%), national (3.2%). Limitations of evidence: Risk of bias due to underreporting of HAIs, methodological heterogeneity, predominance of cross–sectional studies, low use of molecular methods, lack of modeling, and uneven geographical coverage. Overall interpretation and implications: surveillance of HAIs in Africa remains fragmented and poorly standardized. There is a need to strengthen national systems, integrate molecular methods, train professionals, and promote interventional research. The WHO GLASS program can serve as a framework for harmonizing surveillance.

## 1. Introduction

Nosocomial infections (NI) or healthcare–associated infections have become a major global safety concern for both patients and healthcare professionals [1,2,3,4,5]. This term encompasses infections that occur during or after diagnostic, therapeutic, palliative, preventive, or educational care, provided they were neither present nor incubating at the beginning of the care [6].

Nosocomial infections (NI) are infections contracted in hospital. The term ‘healthcare–associated infections’ (HAIs) [7] is now preferred because it includes not only infections contracted in hospital, but also in other healthcare settings (nursing practices, convalescent homes) and even during home care. For ease of reading, the term HAI will be used throughout this article.

HAIs disproportionately affect low– and middle–income countries, where infrastructures are often underdeveloped [8]. Despite global efforts, such as the World Health Organization (WHO) guidelines for infection prevention in hospital practices, implementation remains uneven, particularly in resource–limited settings [8]. Although the public health problem of HAIs in developing countries has been known for decades, these HAIs affect at least 7% of patients in developed countries and 10% in developing countries [9]. Middle–income countries had a national HAIs surveillance system in 12.5% of cases. Most systems operated on a voluntary basis by health services. These systems monitored the incidence of HAIs using the definitions of the Center for Disease Control and Prevention [10]. This paradigm shift has been largely driven by evidence from seminal studies such as the Study on the Efficacy of Nosocomial Infection Control, or SENIC Project, which demonstrated the effectiveness of surveillance and control programs [11].

However, these infections are insufficiently researched, diagnosed, and reported in intertropical Africa [12]. They are a cause for concern due to their high morbidity and mortality rates, and especially due to the emergence of multi–resistant bacteria (MDR).

In intertropical Africa, the risk of HAIs may seem marginal compared to major public health problems such as malnutrition, childhood infections, malaria, AIDS, and violence–related diseases. HAIs are underreported.

The link between HAIs and multidrug–resistant bacteria is an integral part of the HAI problem that can no longer be ignored in Africa [13,14].

Surveillance systems exist in developed countries and provide regular reports on national trends in endemic HAIs [15,16,17,18,19,20]. African countries have established national HAIs surveillance systems, as highlighted in the WHO module on patient safety [21].

Assessing the significance of HAIs in sub–Saharan Africa requires first examining published data. Over two decades, only a few dozen publications have addressed HAIs in patients in intertropical Africa. Most of these are retrospective studies [4].

Previous literature reviews of HAIs in developing countries [22] covered the period between 1995 and 2008, while another, which focused on the World Health Organization (WHO) African Region, covered the period between 1995 and 2009 [23].

These reviews highlighted the need to strengthen frequent surveillance of HAIs. In addition, HAI databases from different healthcare facilities across Africa differ in their methodological approach. Several literature reviews have provided updates on the occurrence of HAIs in Africa from 2010 to 2017, as well as on the contribution of emerging antimicrobial resistance to healthcare delivery in Africa [24].

From 2009 to 2018, a review addressed healthcare–associated bacterial infections in Africa [25]. A systematic review of recent articles published between 2013 and 2016 on antimicrobial resistance in Africa was conducted [26].

An assessment of HAI surveillance in certain African countries was conducted. In Ethiopia, a literature review was conducted by Shiferaw et al. on articles published between 2000 and 2019 on surgical site infections and associated factors [27].

In South Africa, a 10–year literature review was conducted from 2005 to 2014 on bloodstream infections with ESBL–producing and non–ESBL–producing *Escherichia coli* in children in a tertiary referral hospital [28] and then from 2009 to 2013 on exposure to infectious diseases and epidemics in a South African neonatal unit and examination of the epidemiology of neonatal epidemics in Africa [29].

These previous studies reported a wide range of surveillance methods for HAIs–related pathogens. Given the significant differences between the surveillance methods reported in the literature, there is an urgent need for a systematic analysis to synthesize existing knowledge and identify gaps in our understanding.

The main research question addressed is ‘What surveillance tools are used for HAIs in countries in the WHO African Region?’ The secondary questions are ‘What are the types and sources of HAIs?’; ‘How is HAI surveillance organized in the WHO African Region? Which pathogens contribute to these infections and how common are multidrug–resistant (MDR) species in HAIs?’

In this context, the present study aims to describe the organizational methods of surveillance, the pathogens involved, and the frequency of multidrug–resistant HAI species in the WHO African region.

## 2. Materials and Methods

A systematic literature review assessed infection surveillance tools associated with healthcare in WHO African countries. The literature search for this systematic review was conducted in accordance with the PRISMA (Preferred Reporting Items for Systematic Reviews and Meta–Analyses) [30]. All steps of the systematic review were carried out using the free web tool *RAYYAN*.

### 2.1. PICO Elements (Population (Or Patient/Problem), Intervention, Comparison, and Outcome (Result)) [31,32]

Population (P): the target population consists of patients receiving care in any type of health facility (university hospitals, district health centers, clinics, etc.) located in member countries of the WHO African Region.Intervention (I): involved the implementation of an active and continuous surveillance program for HAIs or hospital–acquired infections using epidemiological methods.Comparison (C): the local or national scope of the HAI surveillance methods implemented and the existence of a structured or unstructured surveillance and prevention program.Outcome (O): Reduction in healthcare–associated infections, prevention of antimicrobial resistance, improvement of patient safety.

PICO question: What are the HAI surveillance methods and their scope in WHO African countries compared to limited or no implementation of surveillance to reduce healthcare–associated infections, prevent antimicrobial resistance, and improve safety outcomes?

### 2.2. Search Strategy

For this systematic review, we undertook a literature search and review process according to a protocol designed prior to data collection. The search was conducted between January and March 2025.

To assess methods of HAI surveillance in WHO Afro countries, we extracted studies from *PubMed*, *Web of Science*, *Science Daily*, the Cochrane Database of Systematic Reviews, *African Journals Online Library, and free–text web* searches using *Google Scholar*. Reviewers, KKF, DKM, and GKN independently searched these databases to identify articles published from January 2011 to December 2024. Our search terms included a combination of keywords and MeSH terms (*Medical Subject Headings*) such as “nosocomial infection”, “hospital–acquired infection”, “healthcare–associated infection”, “surveillance”, “Africa”, “individual country name”. Boolean operators (AND/OR) were used to refine and combine terms.

Additionally, we reviewed the references of all included studies to identify further relevant research, ensuring a comprehensive synthesis of available evidence on HAI surveillance methods in WHO Afro countries.

A separate search was conducted in the WHO regional medical database for Africa, African Index Medicus, using a shorter list of essential keywords and time restrictions.

### 2.3. Selection Process

The selection process consisted of a two–stage selection process to compile a list of documents relevant to our research question. Reviewers from the team conducted independent blind checks to ensure objectivity.

First, inclusion and exclusion criteria were defined for the review of titles and abstracts of the identified documents (results of the search in bibliographic databases), which allowed an initial filtering of the literature. In accordance with the PICO principles, the inclusion criteria were population, humans, exposure, healthcare–associated or hospital–acquired infections and the outcome, the HAI surveillance method.

Exclusion criteria were applied articles concerning literature reviews on HAI; articles outside the period and area of study; articles published in languages other than French or English.

If the necessary information in a file was not clearly indicated in the title or abstract, it was selected for the next selection stage.

In the second stage, a full–text selection was carried out for reports (full–text documents) that passed the first filter using additional criteria, full–text accessibility with English or French as the language and reporting an HAIs, accompanied by a description of the methodology used to obtain the data.

In the event of discrepancies between the results obtained by the two reviewers during either screening phase, discussions took place to reach a consensus on the inclusion or exclusion of problematic documents. Following these checks and harmonization, a final list of documents to be extracted was drawn up. The reference lists of the included documents was also reviewed to identify and add relevant records that may have been missed during the initial search. If multiple studies were identified in a single document, they were treated individually during the data extraction phase.

### 2.4. Data Extraction

A data extraction framework was established to facilitate a systematic and efficient analysis of the studies by extracting relevant information. This framework was applied to the selected studies. The categories and labels, designed to be as comprehensive as possible, are presented in Table 1.

### 2.5. Assessment of the Quality or Risk of Bias of Articles

We developed a tool with a series of criteria (Table 2) to identify potential sources of bias, which allowed us to assess the rigor of each study. This tool is an adaptation of the Joanna Briggs Institute (JBI) Critical Appraisal Checklist [30]. The criteria, presented in Table 2, allowed the reviewers to rate the included articles on a scale from low (1 point) to moderate (2 points) or high (3 points). Higher scores indicate that the study had minimal bias, used a sound methodology, and carefully considered potential limitations.

Our tool to assess the overall risk of bias in the included studies considered scales, checklists, and individual components. This tool had scales in which various quality components are rated and combined to give a summary score [32].

The methodological quality assessment scores allowed the articles to be categorized into 3 groups (Table 3).

This tool assesses the methodological rigor and reliability of studies in several areas, including the clarity of inclusion criteria, the appropriateness of the study design, the validity of the outcome measure, and the adequacy of the data analysis. Two reviewers independently conducted the quality assessment, and any disagreements were resolved after consultation with a third reviewer.

To assess the different sources of HAIs, a list of descriptions was selected and detailed in Table 4.

The validated protocol was submitted to the Prospero platform and was registered under number CRD420251032268.

Statistical analysis was performed with Microsoft Excel for all frequency description calculations and graphs. The map was created using QGIS Desktop 3.30.1.

## 3. Results

### 3.1. Document Research, Selection Process, and Characteristics of the Selected Studies

#### 3.1.1. Document Research and Selection Process

The database search yielded 10,362 unique articles (Figure 1). After screening titles and abstracts, 95 articles met the inclusion criteria and were therefore eligible for full–text review. Five [5] additional articles were identified through reference review.

Each included study underwent a risk of bias assessment, which identified potential bias that could affect the reliability of the results. No study had a risk of bias sufficiently high to warrant exclusion from this analysis.

#### 3.1.2. Characteristics of the Selected Studies

In Table 5, this review included 95 observational studies, including 78 cross–sectional studies (82.1%), 10 cohort studies (10.4%), and a single quasi–experimental interventional study of the before–after design. The 78 included detailed descriptions of multiple cases of healthcare ward patients following exposure to a single source of hazard, including contamination by bacterial, viral, or myco–parasitic pathogens. The three case reports and case series documented sporadic incidents where individuals with unique characteristics became ill after being admitted to a healthcare ward. The quasi–experimental study was an evaluation of the effectiveness of a nosocomial infection control program in a Senegalese neonatal unit to reduce nosocomial bacteremia and bacterial resistance conducted by C. Landre–Peigne in 2011 [32].

The use of cross–sectional studies to monitor HAIs in our countries began a long time ago and remains a predominant method, often based on data from documented epidemics of different pathogens. The oldest evaluative interventional study of the “before and after” and “here and elsewhere” type was conducted in 2011 and has not been conducted since 2011 or in any other country, while infection prevention and control programs are underway in all African countries. Case reports and series have been used over time to determine HAIs in real–life conditions.

Among the publications studied, five were produced during an epidemic caused by bacteria [35,77,120,121] and one study was due to a virus [66]. These epidemics were described in two cross–sectional studies, two cohort studies, and one case–control study [34]. The origin of the epidemics was discussed and linked mainly to hygiene, either through interpatient transmission (overcrowding [120,122]) or inappropriate practices by staff. These epidemics mainly involved highly resistant bacteria such as NDM–1–producing bacteria [34] and ESBL [120]. All of these epidemics were brought under control. However, there were no mathematical modeling studies on outbreaks of diseases related to healthcare–associated infections, combined with mathematical tools to compensate for the lack of field data [35,66,77,120,121].

Figure 2 shows the temporal distribution of the different types of studies, from 2011 to 2024. The evolution of the number of publications related to HAIs was sawtooth. The lowest number of articles was in 2021 (2 articles, 2.1%) and the highest number of publications was in 2018 and 2020 (11 articles, 11.6%).

Table 6 presents the number of studies identified on HAIs and the spread of antimicrobial resistance in a nosocomial environment according to the year of publication.

The geographical distribution of data sources (Table 6) was another relevant parameter, as the distribution of HAI prevalence is not uniform across Africa. Thus, the findings of these studies are not always applicable to HAIs present in other regions or countries. In addition, HAI control methods vary across regions, influencing both the hazard and the level of risk. The West Africa, Southern Africa, and East Africa regions accounted for 57 (60.0%), 16 (16.8%), and 9 (9.5%) of the studies, respectively.

The characteristics of the selected studies are summarized in Table 6. This highlights the frequencies of case reports, case series, before–and–after studies, and cross–sectional studies, as well as the regions of origin of these studies, over the following three publication periods: 2011–2015; 2016–2020; 2021–2024.

In all countries, three recorded the highest number of publications on healthcare–associated infections (HAIs). The first country was Nigeria with 16 articles, followed by South Africa (14 publications), and Ghana with 11 publications. Other countries published very few articles over the ten years covered by this study. However, even the leading countries have an average of only 1 to 1.8 publications per year (Figure 3).

When analyzing the sources of HAIs, Table 7 shows that the main sources of contamination were exogenous. The microorganism comes from a source external to the patient in 79 articles (83.2%). Some infections are both endogenous (the index patient was infected) and exogenous (transmission to other patients and caregivers) in 8.4% of cases.

Figure 4 presents a summary analysis of the origins of HAIs in our literature review. With nearly half of the references (43.2%), HAIs remain the central concern in the literature, reflecting their high incidence and the complexity of their prevention in the intraoperative setting. The burden of invasive infections represented by bacteremia/bloodstream infections (35.8%) and respiratory infections (27.4%) is significant, often associated with morbidity.

Table 8 summarizes the results of the risk of bias assessment of the selected articles. The articles had excellent methodological rigor (85.2%) with an overall low risk of bias.

### 3.2. Organization of HAIs Surveillance

The analysis of the scope of the studies shows that 96.8% of surveillance interventions have a local scope (92 studies), while 3 studies have a national scope (3.2%).

Only three countries (Ghana, South Africa, and Benin) reported studies based on national data. These studies used cross–sectional methods [80,92,101]. The vast majority of studies were local in scope, employing various methods. Among these methods, cross–sectional studies were widely reported in 78 articles, representing 82.1% (Table 9).

A certain number of studies (94.7%, 90 studies) specified the laboratory investigation technique for HAIs samples. Among the publications that did so, only two studies (2.1%) explored HAIs samples using a genotypic technique and two studies with genomics (2.1%). Regarding the methods, 67 studies used purely phenotypic methods (70.5%) compared to 16 studies that used both phenotypic and genotypic methods (16.8%). National studies conducted in three African countries used phenotyping as the method for investigating HAIs samples in the laboratory (Table 10).

### 3.3. Reported Pathogens and Resistance Profile

#### 3.3.1. Reported Pathogens

Table 11 shows that in the 95 publications the vast majority of reported HAIs concerned the 48 bacterial species (50.5%). Only 4 species of *Candida fungi* (4.2%) were reported, and 4 species of viruses (4.2%) were reported.

The top five most monitored bacteria in the articles were *Staphylococcus aureus* in 51 articles (53.7%), *Escherichia coli* in 41 articles (43.2%), *Pseudomonas aeruginosa* in 35 publications (36.8%), *Klebsiella pneumoniae* was concerned by 31 articles (32.6%), and finally *Acinetobacter baumannii* cited in 21 articles (22.1%).

Although HAIs are often associated with antibiotic–resistant bacteria, eight articles (8.4%) did not specify a particular pathogen.

#### 3.3.2. Resistance Profile

Table 12 shows that the ESBL bacterial phenotype was the most frequently reported profile (26 references, 27.4%) and the MRSA profile came second (20 references, 21.1%). Multidrug resistance (MDR) was also well documented with thirteen references, 13.7%. Resistance to carbapenems and methicillin is well represented. Many specific molecular profiles were identified but with fewer references. An important category concerns unspecified or unstudied resistance (twenty–five references).

Table 13 shows very high rates of resistance of the usual isolates in the studies. *Escherichia coli* showed a rate of 97.6% against ampicillin. For *Acinetobacter baumannii*, the rate of MDR varied from 62.1 to 90%, and for carbapenems, it was from 47.6 to 75.3%. All the listed germs are MDR.

Apart from resistance phenotypes, only 11 articles out of 95 reported cases of resistance genotypes. The resistance genotypes most reported in this study concerned beta–lactams, fluoroquinolones, and aminoglycosides, respectively. These genotypes were represented by the CTX–M–15 gene in 2 articles [47,108], qnr (*qnrB*, *qnrS*) and AAC(6′)–Ib–cr in 1 article [108], rendering several families of antibiotics ineffective. Other resistance genes were also reported, including the mecA gene (MRSA, methicillin resistance) in one article [74] and VanA in 1 article [60].

## 4. Discussion

### 4.1. Summary of the Main Results

This systematic review aimed to identify healthcare–associated infection (HAI) surveillance tools in the WHO African Region. A total of 95 studies were included, predominantly cross–sectional (82.1%) and cohort (10.5%) studies. Most studies were from West Africa (60%), with Nigeria, South Africa, and Ghana as the main contributors. The most frequently reported pathogens were *Staphylococcus aureus* (53.7%), *Escherichia coli* (43.2%), and *Pseudomonas aeruginosa* (36.8%). The main resistance patterns were ESBL (27.4%) and MRSA (21.1%). The majority of studies (96.8%) were local in scope and used phenotypic methods (70.5%), with only a few studies incorporating genotypic or genomic approaches. This significant variability in the types of studies and methodologies employed included differences in the populations exposed, the pathogens incriminated, and the laboratory investigation techniques.

Our results are broadly consistent with those of other recent systematic reviews of HAIs in Africa, but some methodological and contextual differences are worth highlighting.

In a review Irek et al. also identified a predominance of cross–sectional studies and underreporting of HAIs in sub–Saharan Africa. However, she highlighted a lack of data on fungal and viral infections, which our study confirms (only 4.2% for each category) [22].

In their study, Talaat et al. highlighted the emergence of multidrug–resistant bacteria (MDR) in HAIs in Africa, with an increasing prevalence of ESBL and carbapenem–resistant Enterobacteriaceae. Our results are consistent, with 27.4% of studies reporting ESBL and 10.5% CRE [119].

The Global Report on Infection Prevention and Control 2024 [122] assessed HAI surveillance systems in referral hospitals in Africa and found that less than 30% of countries had national surveillance systems. Our findings go even further: only three studies (3.2%) were national in scope, highlighting the critical lack of harmonized data at the national level.

During our literature search, we identified several studies reporting HAIs related to case series and surgical site infections. These studies provide valuable data on contamination levels in different matrices [38,41].

The most frequently studied pathogens represented in the literature were bacteria. The most isolated bacterial species were *Escherichia coli*, *Staphylococcus aureus*, and *Klebsiella pneumoniae.*

HAI surveillance in the WHO African Region represents a major public health challenge, as demonstrated by this systematic review covering the period from 2011 to 2024. The results highlight the methodological, epidemiological, and structural challenges faced by African countries in HAI surveillance, as well as the progress made and persistent gaps, as well as opportunities to improve HAI surveillance and prevention in the African region.

### 4.2. Current Status of HAI Surveillance in Africa

The results of this review showed marked heterogeneity in HAI surveillance systems across the African region. Although a few countries, such as Nigeria, South Africa, and Ghana, have published a relatively high number of HAI studies, the majority of African countries remain underrepresented in the scientific literature. This disparity likely reflects differences in the infrastructural capacity, financial resources, and technical expertise available to conduct robust HAIs studies.

Most of the included studies (82.1%) were cross–sectional in nature, suggesting a predominance of one–off surveys rather than continuous surveillance systems. Only a few longitudinal or cohort studies were identified, limiting the ability to assess temporal trends in HAIs. Furthermore, only one interventional “before–after” study was identified, highlighting the lack of impact evaluations of HAI prevention programs in the region, where they exist.

The scope of the studies was predominantly local (96.8%), with very little national or regional data. This focus on specific contexts, while valuable for understanding local dynamics, makes it difficult to generalize the results and implement coordinated strategies on a larger scale.

### 4.3. Heterogeneity of Monitoring Methods

This review identified methodological diversity in HAI studies, with a predominance of cross–sectional studies (82.1%), followed by cohort studies (10.5%), and case reports (3.2%). This heterogeneity reflects the logistical and financial constraints of African health systems, where one–off cross–sectional studies are often preferred due to their feasibility and low cost. However, these studies offer a limited view over time and do not allow monitoring the evolution of HAIs or assessing the real impact of interventions in the long term. Only one quasi–experimental “before–after” study was identified, conducted in Senegal in 2011, highlighting the glaring lack of intervention evaluations in the region [32].

Geographic disparities are also notable, with studies concentrated in West Africa (60.0%) and Southern Africa (16.8%), while Central and Northern Africa are underrepresented. Nigeria, South Africa, and Ghana dominate publications, perhaps reflecting more developed research capacities in these countries. These disparities highlight the need to strengthen surveillance capacities in less–covered regions to obtain a more balanced view of the HAI situation at the continental level.

### 4.4. Pathogens Involved in HAIs and Predominance of Multidrug–Resistant Pathogens

The most frequently reported pathogens in HAIs include *Staphylococcus aureus* (51 studies), *Escherichia coli* (41 studies), and *Klebsiella pneumoniae* (31 studies). These bacteria, often associated with antibiotic resistance profiles (ESBL, MRSA, CRE), pose a major challenge for the management of infections in hospital settings. The presence of *Acinetobacter baumannii* and *Pseudomonas aeruginosa*, also documented, worsens this picture due to their resistance to last–resort antibiotics [23].

Monkeypox virus (one study) and Lassa fever virus in the context of HAIs are a reminder that threats are not limited to bacteria. The COVID-19 pandemic has highlighted the risks of HAIs transmission of viruses, particularly in healthcare settings with limited resources [20,124]. These findings highlight the importance of integrated surveillance, covering bacteria, viruses, and other pathogens.

The most frequently cited sources of HAIs were invasive medical devices and the hospital environment. These findings highlight the need to strengthen hygiene practices, equipment sterilization, and staff training to reduce the risk of HAIs.

### 4.5. Laboratory Methods and Technical Limitations

Of the studies analyzed, 93.4% specified the laboratory methods used to identify pathogens. However, the majority (78 studies) used phenotypic techniques, while only 3 studies used genotypic methods, and 2 used genomic approaches. This predominance of phenotypic methods, although more accessible and less expensive, limits the ability to accurately characterize resistance mechanisms and detect outbreaks linked to specific species.

Genotypic and genomic methods, although rarer, offer significant advantages for HAI surveillance, including better resolution for species typing and detection of resistance genes. Their wider adoption in Africa could improve the quality of surveillance data and facilitate international comparisons.

Limitations of the evidence presented in the analysis: The identification of few eligible studies or studies involving small numbers of participants, leading to imprecise estimates; the risk of bias in studies or missing results; or identifying studies that only partially or indirectly answer the review question, raising doubts about their relevance and applicability to particular patients, settings, or other target audiences.

Limitations of the review processes used: The decision to restrict inclusion to studies in English and French only, to search only a limited number of databases, to entrust file review or data collection to a single reviewer, or not to contact study authors to clarify ambiguous information. We were probably unable to access all potentially eligible study reports or conduct some of the planned analyses due to insufficient data.

Despite the efforts described in this review, several limitations persist in the current monitoring system:

Limited scope of studies: The majority of studies (96.2%) are local in scope, focusing on individual hospitals or specific regions. Only three studies are national in scope, and only one is regional. This fragmentation limits the generalizability of results and the implementation of coordinated policies at the continental level.

Laboratory methods: Although 93.4% of studies specify investigation techniques, only 3% use genotypic or genomic methods, which are essential for understanding epidemic dynamics and antibiotic resistance. The majority of studies rely on phenotypic methods, which are less precise for identifying resistance mechanisms [21].

Underreporting and HAIs: As several authors point out, HAIs are often underreported in sub–Saharan Africa due to the lack of standardized surveillance systems, low awareness among health professionals, and diagnostic constraints [7]. This underreporting distorts the true estimate of the HAI burden and hinders prevention efforts.

Lack of modeling: No mathematical modeling studies were identified, although these tools could compensate for the lack of field data and anticipate epidemiological trends [22].

### 4.6. Recommendation to Public Health Policies and Professionals

The results of this review highlight the urgency of strengthening HAI surveillance systems in Africa. The following recommendations could guide policymakers and health professionals.

Develop national surveillance systems: African countries should invest in continuous and standardized surveillance systems, aligned with WHO guidelines, to generate comparable and actionable data.

Promote the use of molecular methods: The integration of genotypic and genomic techniques in reference laboratories would enable more accurate monitoring of pathogens and AMR.

Strengthening local capacity: Targeted training for health personnel and laboratory technicians is needed to improve data quality and the implementation of prevention protocols.

Improve hygiene and infection control practices: Interventions such as hand hygiene promotion, equipment sterilization, and medical waste management could significantly reduce the incidence of HAIs.

Encourage intervention research: More “before–and–after” or randomized studies are needed to evaluate the effectiveness of HAI prevention programs.

### 4.7. Recommendations for GLASS

Following this literature review on HAIs in the WHO Afro region, it is recommended that a laboratory surveillance program for AMR–related HAIs be strengthened in countries of the WHO Afro region. For a national laboratory surveillance program, government commitment and support are non–negotiable. GLASS should also encourage the establishment of supranational coordination centers that will coordinate national reference centers, which in turn will systematically collect, analyze, and share AMR–related HAI data nationally and internationally. These regional reference laboratories will provide technical support, including training, capacity building, and policy advice.

## 5. Conclusions

This systematic review provides an overview of the challenges and opportunities related to HAI surveillance in Africa. Although progress has been made in publishing HAI surveillance data, particularly in leading countries such as Nigeria and South Africa, significant gaps persist in terms of geographical coverage, surveillance methods, and access to advanced technologies. A coordinated approach, involving governments, health institutions, and international partners, is essential to improve HAI surveillance and prevention in the region. By addressing these gaps, Africa can better protect the health of patients and healthcare workers, while contributing to the global fight against antimicrobial resistance.

The capacity of countries in the WHO Afro region varies greatly in the area of combating AMR and consequently AMR related to HAIs. However, this capacity has been significantly hampered by the availability of capacity building and, more recently, by the emergence of the COVID-19 pandemic, which has also been added to the long list of healthcare–associated infections.

Thus, the true burden of HAIs in this systematic review is underreported and is perhaps greater in countries with weaker health infrastructure. However, this trend is evolving with the growing commitment in Africa to address the global threat of AMR. National technical capacities with the support of partners are now evolving not only to develop national AMR action plans but also to institute national AMR surveillance. The WHO Global Antimicrobial Surveillance System (GLASS) provides a tool to standardize data collection, sharing, and analysis across participating institutions and countries globally to monitor trends and implement controls.

Data on the prevalence of antimicrobial resistance and HAIs in developing countries in African WHO Afro are scarce and unsystematic; thus, authors suggest that intensive survey and surveillance are warranted.

## Figures and Tables

**Figure 1 antibiotics-14-01287-f001:**
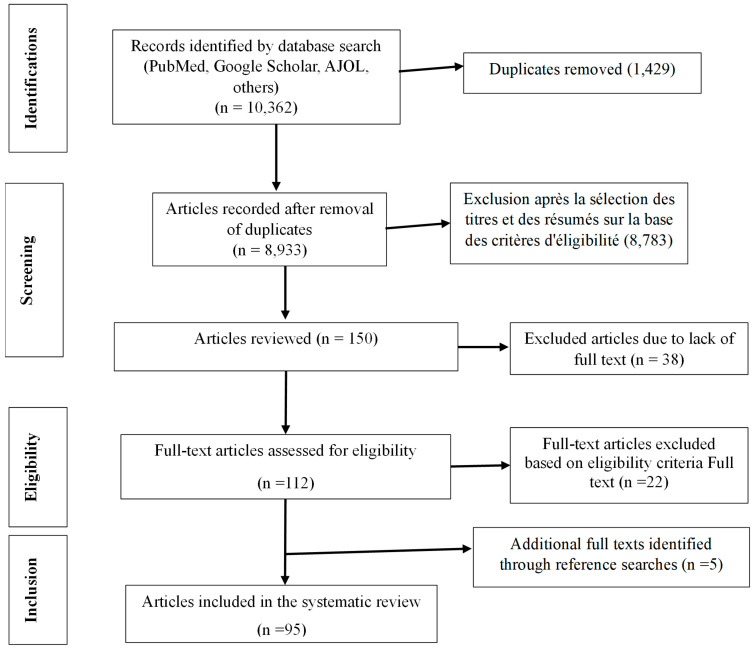
Summary of the selection of studies included in the review according to PRISMA guidelines.

**Figure 2 antibiotics-14-01287-f002:**
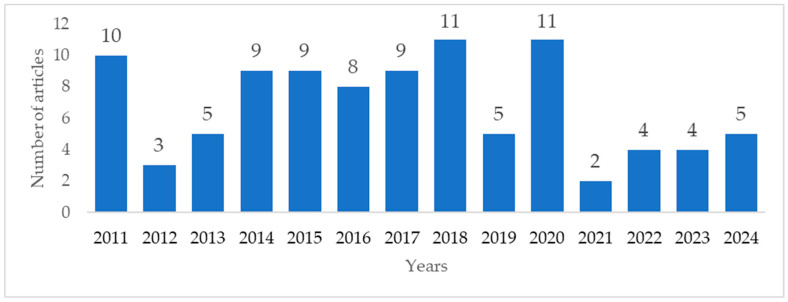
Number of publications on HAIs over time (2011–2024).

**Figure 3 antibiotics-14-01287-f003:**
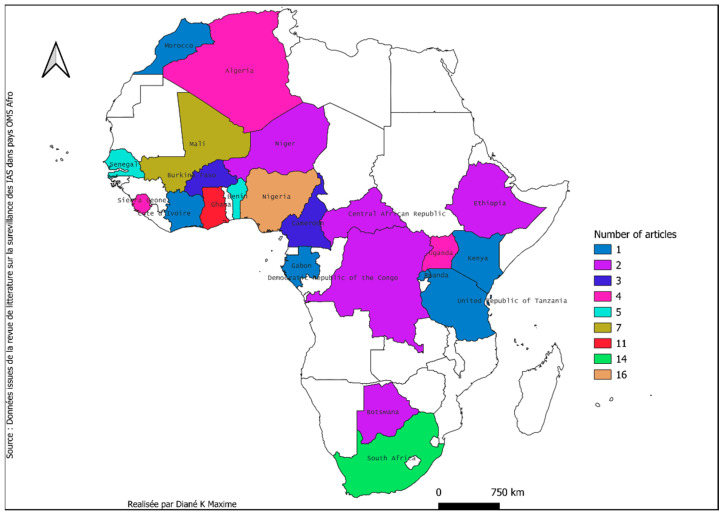
Number of publications on HAIs per countries over time (2011–2024).

**Figure 4 antibiotics-14-01287-f004:**
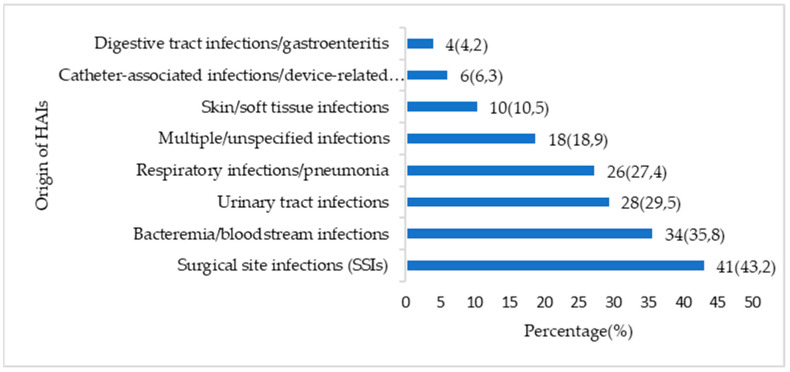
Distribution of healthcare–associated infections by origin.

**Table 1 antibiotics-14-01287-t001:** Data extraction categories, labels, and information to be collected.

Category	Label	Information to Be Collected
General information	Item ID	Awarded by team
	Authors	Authors’ names
	Year of publication	Year of publication of the study
	Title	Article Title
	DOI/PMID/Link	Study ID
Study characteristics	Type of study	Case–control, Descriptive cohort, Before–after study; Case report; Case series Cross–sectional study
	Objectives of the study	Describe the objective of the study
	WHO Region	Region of the world defined by the WHO
	Country	Countries where the study was conducted
	Study period	Duration of the study
Population details	Population or subject studied	General population, children, elderly, etc.
	Sources of infection	Acquired in hospital
	Population information	Gender, age, health status, inpatient/outpatient service, etc.
Exhibition details	Mode of transmission	Individual cases of illness, epidemics, etc.
	HAIs source	Patients, staff; equipment
	Type of samples	Pus swab, urine, blood, catheters, sputum, nasopharyngeal aspirate, expectorations, and hospital consumables, environmental swab
	Number of samples	Number
Laboratory methods	Pathogens identified	Genera and species
	Type of pathogens	Bacteria, fungi, viruses
	Laboratory identification method	Genomic, genotypic, unspecified, phenotypic
	Antibiogram	Antibiotics tested
	AST Guidelines	CLSI; EUCAST–CASFM
	Resistance profile	ESBL, CRAB; CRE; MRSA
Main findings and results	Main results	Key findings of the article
	Main conclusions	Conclusions drawn by the authors
	Additional notes	Any other relevant information

**Table 2 antibiotics-14-01287-t002:** Criteria used to assess the risk of bias in the selected articles.

Criteria	Source of Bias	Applied Scale
Study using data from one’s own work or another source	Data source	1—External; 2—Mixed; 3—Internal
Data used clearly indicated (with source if necessary)	Data availability	1—No; 2—Unclear; 3—Yes
Estimate or measure the size of the exposed population	Size of exposed population	1—Unclear; 2—Estimated; 3—Measured
Estimation or measurement of the size of the sick population	Size of the sick population	1—Unclear; 2—Estimated; 3—Measured
Characteristics of individuals clearly presented	Population health status	1—Not described; 2—Described
Assessment of population size with critical evaluation by the author	Population size	1—No; 2—Yes
Assessment of the author’s intervention method	Type of study	1—Not described; 2—Described
Appreciate the different sources of HAIs	HAIs Sources	1—Not described; 2—Described
Clearly identified HAIs source	HAIs Sources	1—No; 2—Unclear; 3—Yes
Critical evaluation by the authors of the study results and comparison with other studies	Discussion of results	1—No critical evaluation by the authors; 2—Critical evaluation by the authors
Critical assessment by the authors of factors that may influence their results	Discussion of factors influencing the results	1—No critical evaluation by the authors; 2—Critical evaluation by the authors

**Table 3 antibiotics-14-01287-t003:** Different score groups of methodological quality assessment.

Band	Methodological Quality	Score (%)
I	Forte	(28–33) 75–100
II	Moderate	(17–27) 50–75
III	Weak	(<16) <50

**Table 4 antibiotics-14-01287-t004:** Description of HAIs sources.

Source Category	Origin of the Microorganism	Specific Sources	Mode of Transmission Predominant
1. Endogenous Source (Self–contamination)	The patient himself	Patient flora (digestive, cutaneous, respiratory) which enters through an artificial or natural entry point.	Migration/Direct inoculation (during an invasive or surgical procedure): the patient’s germs are introduced into a sterile site (e.g., catheter, surgical site).
2. Exogenous Source (Cross–Contamination)	External to the patient		
	Humans: Other Patients	Infected patients or asymptomatic carriers (especially MDR).	Direct or indirect contact (through the hands of staff or shared equipment).
	Humans: Healthcare Staff	Asymptomatic carrier staff members. Lack of hygiene (hands, clothing).	Handling (hands not disinfected between patients) or airborne (in case of unmasked respiratory infection).
	Environmental/Material: Equipment and Devices	Invasive devices (catheters, probes, ventilators, etc.). Improperly sterilized surgical instruments. Shared surfaces (bed, table, handles) and objects.	Direct inoculation (via the device) or indirect contact (via surfaces/materials).
	Environmental/Material: Inanimate Environment	Water (taps, showers, air conditioning systems). Air (works, ventilation). Food (very rare).	Airborne (inhalation) or waterborne (ingestion, nebulization).

**Table 5 antibiotics-14-01287-t005:** Typology of selected studies.

Type of Study	Bibliographic References	*n* (%)
Case–control	[33,34]	2 (2.1)
Cohort	[35,36]	10 (10.4)
Before–and–after study	[37]	1 (1.1)
Case report	[38,39,40]	3 (3.2)
Case series	[41]	1 (1.1)
Transversal	[42,43,44,45,46,47,48,49,50,51,52,53,54,55,56,57,58,59,60,61,62,63,64,65,66,67,68,69,70,71,72,73,74,75,76,77,78,79,80,81,82,83,84,85,86,87,88,89,90,91,92,93,94,95,96,97,98,99,100,101,102,103,104,105,106,107,108,109,110,111,112,113,114,115,116,117,118,119]	78 (82.1)
Total		95

**Table 6 antibiotics-14-01287-t006:** Temporal and Afro–regional distribution by study types of HAIs.

	Case–Control	Descriptive Cohort	Before-and-After Study	Cross-Sectional Study	Case Report	Case Series	Total n (%)
Years							
2011–2015	2	2	1	29	2	0	36 (37.9)
2016–2020	0	3	0	39	1	1	44 (46.3)
2021–2024	0	5	0	10	0	0	15 (15.8)
African regions							
West Africa	0	7	1	45	3	1	57 (60.0)
Central Africa	0	1	0	7	0	0	8 (8.4)
East Africa	0	0	0	9	0	0	9 (9.5)
South Africa	2	2	0	12	0	0	16 (16.8)
North Africa	0	0	0	5	0	0	5 (5.3)
Total	2	10	1	78	3	1	95

**Table 7 antibiotics-14-01287-t007:** Distribution of types of HAIs according to source.

Type of HAIs Sources	Bibliographic References	n (%)
Endogenous	[44,47,50,57,71,79,90,113]	8 (8.4)
Exogenous	[33,34,35,36,37,38,39,40,41,42,43,45,46,48,49,52,53,55,56,58,59,60,61,62,63,64,65,67,68,69,70,72,73,74,75,76,77,78,80,83,84,86,87,88,89,91,92,93,94,95,96,97,98,100,101,102,103,104,105,106,107,108,109,110,111,112,116,117,118,120,121,122,123,124,125,126,127]	79 (83.2)
Exogenous and Endogenous	[54,66,81,82,85,99,114,115]	8 (8.4)
Total		95

**Table 8 antibiotics-14-01287-t008:** Risk of bias assessments for each included study.

Methodological Quality	Bibliographic References	n (%)
High (>75%)	[33,34,35,36,37,40,41,42,43,45,46,47,48,49,50,51,52,53,54,55,57,58,59,60,61,63,64,65,66,67,69,71,75,76,77,78,79,80,81,82,86,88,89,90,91,92,93,94,95,96,97,98,99,100,101,102,103,104,106,107,108,109,111,112,113,114,115,116,117,118,120,121,122,123,124,125,126,127]	81 (85.2)
Moderate (50–75%)	[38,39,44,56,62,68,70,73,74,83,87,105,110]	13 (13.7)
Acceptable (<50%)	[72]	1 (1.1)

**Table 9 antibiotics-14-01287-t009:** Typical distribution and scope of HAIS studies by country.

Type of Study	Scope	Country	Bibliographic Reference	n (%)
Case–control n = 2 (2.1)	Local n = 2 (2.1)	South Africa	[34,35]	2 (2.1)
Descriptive cohort n = 10 (10.5)	Local n = 10 (10.5)	South Africa	[120,122]	2 (2.1)
		Cameroon	[126]	1 (1.1)
		Ivory Coast	[125]	1 (1.1)
		Ghana	[123]	1 (1.1)
		Mali	[121]	1 (1.1)
		Niger	[124]	1 (1.1)
		Sierra Leone	[36,37,127]	3 (3.2)
Before–after study n = 1 (1.1)	Local n = 1 (1.1)	Senegal	[33]	1 (1.1)
Cross–sectional study n = 78 (82.1)	Local n = 75 (78.9)	South Africa	[26,51,58,64,94,97,99,100,102]	9 (9.5)
		Algeria	[48,60,72,113]	4 (4.2)
		Benin	[45,75,82,90]	4 (4.2)
		Botswana	[117]	1 (1.1)
		Botswana and South Africa	[96]	1 (1.1)
		Burkina Faso	[52,65,76]	3 (3.2)
		Cameroon	[54,61]	2 (2.1)
		Ethiopia	[62,117]	2 (2.1)
		Gabon	[67]	1 (1.1)
		Gambia	[74,77]	2 (2.1)
		Ghana	[47,59,70,78,83,84,85,87,104]	9 (9.5)
		Kenya	[57]	1 (1.1)
		Mali	[86,89,110,112,115]	5 (5.3)
		Morocco	[115]	1 (1.1)
		Niger	[71]	1 (1.1)
		Nigeria	[42,44,46,50,53,55,68,69,73,79,81,87,95,98,106,109]	16 (16.8)
		Uganda	[63,105,107,111]	4 (4.2)
		Central African Republic	[66,109]	2 (2.1)
		Democratic Republic of Congo (DRC)	[93,103]	2 (2.1)
		Rwanda	[49]	1 (1.1)
		Senegal	[88,91]	2 (2.1)
		Sierra Leone	[43]	1 (1.1)
		Tanzania	[56]	1 (1.1)
	National n = 3 (3.2)	South Africa	[101]	1 (1.1)
		Benin	[92]	1 (1.1)
		Ghana	[80]	1 (1.1)
Case report n = 3 (3.2)	Local n = 3 (3.2)	Mali	[38]	1 (1.1)
		Senegal	[39,40]	2 (2.1)
Case series n = 1 (1.1)	Local n = 1 (1.1)	Gambia	[41]	1 (1.1)
Total				95 (100.0)

**Table 10 antibiotics-14-01287-t010:** Distribution of study scopes of HAIs investigation methods in the HAI laboratory.

Scope n (%)	Investigation Methods n (%)	Country	Bibliographic Reference	n (%)
Local n = 92 (96.8)	Clinical n = 5 (5.3)	South Africa	[58]	1 (1.1)
		Ghana	[78]	1 (1.1)
		Uganda	[63,105]	2 (2.1)
		Sierra Leone	[37]	1 (1.1)
	Genomics n = 2	Gambia	[74]	1 (1.1)
		Ghana	[87]	1 (1.1)
	Genotypic n = 2	Nigeria	[98]	1 (1.1)
		Central African Republic	[66]	1 (1.1)
	Phenotypic n = 67 (70.5)	South Africa	[26,35,51,64,94,97,99,102,122]	9 (9.5)
		Algeria	[48,72,113]	3 (3.2)
		Benin	[45,90]	2 (2.1)
		Botswana	[117]	1 (1.1)
		Botswana and South Africa	[96]	1 (1.1)
		Burkina Faso	[52,65,76]	3 (3.2)
		Cameroon	[54,61,126]	3 (3.2)
		Ivory Coast	[125]	1 (1.1)
		Ethiopia	[62,117]	2 (2.1)
		Gabon	[67]	1 (1.1)
		Gambia	[41,77]	2 (2.1)
		Ghana	[84,104]	2 (2.1)
		Kenya	[57]	1 (1.1)
		Mali	[38,86,89,110,112,116,121]	7 (7.4)
		Morocco	[115]	1 (1.1)
		Niger	[71,124]	2 (2.1)
		Nigeria	[42,44,50,53,55,68,69,79,81,86,95,106,109]	13 (13.7)
		Uganda	[107,112]	2 (2.1)
		Central African Republic	[108]	1 (1.1)
		Democratic Republic of Congo (DRC)	[93,103]	2(2.1)
		Rwanda	[49]	1 (1.1)
		Senegal	[33,88,91]	3 (3.2)
		Sierra Leone	[36,43,127]	3 (3.2)
		Tanzania	[56]	1 (1.1)
	Phenotypic + genotypic n = 16 (16.8)	South Africa	[34,100,120]	3 (3.2)
		Algeria	[60]	1 (1.1)
		Benin	[75,82]	2 (2.1)
		Ghana	[47,59,70,83,85,123]	6 (6.3)
		Nigeria	[46,73]	2 (2.1)
		Senegal	[39,40]	2 (2.1)
National n = 3 (3.2)	Phenotypic n = 3 (3.2)	South Africa	[101]	1 (1.1)
		Benin	[92]	1 (1.1)
		Ghana	[80]	1 (1.1)
Total				95

**Table 11 antibiotics-14-01287-t011:** Distribution of pathogens involved in HAIs studies.

Name of the Pathogen	Bibliographic References	n (%)
Bacteria n = 48		(50.5)
*Acinetobacter baumannii*	[26,33,35,43,45,48,51,56,83,86,90,91,92,93,96,100,106,114,116,123,124]	21 (22.1)
*Acinetobacter* spp.	[67,70,79,85,90,91,112,116,126]	9 (9.5)
*Alcaligenes*	[69]	1 (1.1)
*Bacillus* spp.	[104]	1 (1.1)
Gram–negative bacteria	[53,88,110,114,121,127]	6 (6.3)
Gram–positive bacteria	[53,88,127]	3 (3.2)
*Bacteroides* spp.	[69,79]	2 (2.1)
*Bacteroides fragile*	[42]	1 (1.1)
*Citrobacter* spp.	[62,85,92,106,126]	5 (5.3)
*Citrobacter friends*	[93]	1 (1.1)
*Clostridium* spp.	[69]	1 (1.1)
*Coagulase–negative Staphylococcus*	[91,92,117]	3 (3.2)
*Corynebacterium aurimucosum*	[39]	1 (1.1)
*Enterobacter* spp.	[61,62,67,85,106,116,126]	7 (7.4)
*Enterobacter cloacae*	[33,34,45,51,54,61,86,116,125]	9 (9.5)
*Enterobacter faecalis*	[86]	1 (1.1)
*Enterococcus faecalis*	[43,56,93,94,124]	4 (4.2)
*Enterococcus faecium van A*	[60]	1 (1.1)
*Enterococcus* spp.	[36,67,68,92]	4 (4.2)
*Escherichia coli*	[36,42,44,45,46,47,48,49,52,54,55,56,57,59,61,65,67,68,71,76,80,82,84,85,86,89,90,92,93,95,101,103,104,106,111,112,116,122,123,124,126]	41 (43.2)
*Hafnia alvei*	[46,106]	2 (2.1)
*Klebsiella oxytoca*	[36,46]	2 (2.1)
*Klebsiella pneumoniae*	[33,34,36,42,43,45,48,49,51,54,56,57,61,65,67,68,71,72,84,86,89,90,95,96,97,99,106,112,116,123,124,125]	31 (32.6)
*Klebsiella* spp.	[53,58,62,63,81,85,86,88,91,97,108,111,113,114,127]	15 (15.8)
*Listeria monocytogenes*	[96]	1 (1.1)
*Morganella*	[84]	1 (1.1)
*Peptococcus* spp.	[42]	1 (1.1)
*Proteus mirabilis*	[36,42,46,54,56,61,86,106,113]	9 (9.5)
*Proteus* spp.	[55,62,68,79,87,95,106]	7 (7.4)
*Proteus vulgaris*	[126]	1 (1.1)
*Providencia* spp.	[62,69]	2 (2.1)
*Providencia stuartii*	[56]	1 (1.1)
*Pseudomonas aeruginosa*	[36,42,43,44,45,46,48,51,54,56,57,61,62,68,71,76,79,80,84,85,86,89,90,91,92,93,95,102,103,104,106,112,113,115,116,123]	35 (36.8)
*Pseudomonas* spp.	[102,111,126]	3 (3.2)
*Salmonella arizonae*	[126]	1 (1.1)
*Salmonella enterica*	[40,120]	2 (2.1)
*Non–typhoidal Salmonella*	[124]	1 (1.1)
*Salmonella* spp.	[92,103]	2 (2.1)
*Salmonella typhi*	[103]	1 (1.1)
*Serratia liquefaciens*	[77]	1 (1.1)
*Serratia marcescens*	[34,61]	2 (2.1)
*Shigella* spp.	[103]	1 (1.1)
*Staphylococcus aureus*	[33,36,42,43,44,45,48,49,51,52,54,55,56,57,59,61,62,64,67,68,69,71,72,73,74,75,76,80,86,87,89,90,91,92,93,95,96,97,99,101,102,103,104,106,111,112,115,117,123]	51 (53.7)
*Staphylococcus haemolyticus*	[59]	1 (1.1)
*Staphylococcus* spp.	[52,56]	2 (2.1)
*Group B Streptococcus*	[96,101]	2 (2.1)
*Streptococcus* spp.	[69,91]	2 (2.1)
*Streptococcus viridans*	[69]	1 (1.1)
Fungi/mycoses n = 4		(4.2)
*Candida albicans*	[26,51,56,86,92,101,113,115,122]	9 (9.5)
*Candida krusei*	[122]	1 (1.1)
*Candida parapsilosis*	[122]	1 (1.1)
*Candida* spp.	[42,44,52,53,81,92,97,99,113,116]	11 (11.6)
Virus n = 4	[4]	4 (4.2)
Virus (*HVA*, *HVB*, *HVC*)	[72]	1 (1.1)
Virus (*RSV*, *Adenovirus*)	[97,99]	2 (2.1)
*Lassa fever virus*	[98]	1 (1.1)
*Monkeypox virus* (Mpox)	[66]	1 (1.1)
UNSPECIFIED n = 9	[9]	(9.5)
Not specified	[37,41,50,58,63,78,94,105,107]	9 (9.5)

NB: Individual studies have reported several agents.

**Table 12 antibiotics-14-01287-t012:** Characteristics of phenotypic resistance profiles of bacteria in the articles studied.

Phenotypic Profile	Bibliographic References	Number n = 95 (%)
ESBL	[26,43,45,48,56,62,65,67,76,80,85,86,89,90,91,96,98,100,102,107,120,121,123,124,125,127]	26 (27.4)
MRSA/Methicillin–resistant	[33,48,53,56,62,64,67,73,74,75,88,90,92,95,97,99,101,111,113,123]	20 (21.1)
MDR/Multi–resistance	[53,54,56,60,61,62,70,84,100,111,117,120,126]	13 (13.7)
CRE/Carbapenemases	[35,36,45,80,85,96,100,123,124,127]	10 (10.5)
C3GR	[33,38,80,85,95,123]	6 (6.3)
Tetracycline R	[44,59,68,84,86,87]	6 (6.3)
Beta–lactamase	[44,68,89,125]	4 (4.2)
Ceftazidime R	[88,92,115]	3 (3.2)
Fluconazole R	[97,122]	2 (2.1)
Cephalosporinase	[110]	1 (1.1)
Amphotericin B (sensitive)	[122]	1 (1.1)
CRAB	[26,48,100]	3 (3.2)
VRE	[92,113]	2 (2.1)
Partial resistance/not specified	[34,35,39,41,49,50,55,58,61,63,69,72,77,78,79,87,94,98,102,103,104,105,107,109]	25 (26.3)

ESBL: Extended–spectrum beta–lactamase–producing enterobacteria; MRSA: Methicillin–Resistant Staphylococcus aureus; MDR: Multidrug–Resistant Bacteria; CRE: Carbapenem–Resistant Enterobacteriaceae; C3GR: Resistance to third–generation cephalosporins; CRAB: Carbapenem–Resistant *Acinetobacter baumannii*; R: Resistance; VRE: vancomycin–resistant enterococcus.

**Table 13 antibiotics-14-01287-t013:** Resistance rates and phenotypic profiles of bacteria to antibiotics in articles.

Name of the Pathogen	Resistance Rate (%)	Common Resistance Phenotypes	Bibliographic References
*Escherichia coli*	ESBL (56–79.2)	ESBL, resistance to 3rd generation cephalosporins, multi–resistance	[36,65,82,84,111]
	Ampicillin resistance (97.6)		
	Amoxicillin resistance (95.2)		
*Klebsiella pneumoniae*	ESBL (59.2–80.3)	ESBL, CRE, resistance to beta–lactams, cephalosporins	[26,38,72,85,96,124]
	Carbapenem resistance (1.3–5.24)		
*Staphylococcus aureus*	MRSA (15–100)	MRSA, oxacillin resistance, multi–drug resistance to beta–lactams	[26,48,62,64,75,87]
	Penicillin resistance (88.6–100)		
*Pseudomonas aeruginosa*	Ciprofloxacin resistance (50–68.2)	Resistance to fluoroquinolones, cephalosporins, carbapenems	[35,48,76,91,123]
	Resistance to meropenem (31)		
*Acinetobacter baumannii*	Multidrug resistance (MDR) (62.1–90)	CRAB, resistance to cephalosporins, carbapenems, colistin (rare)	[26,35,48,70,83,100]
	Carbapenem resistance (47.6–75.3)		
*Enterobacter* spp.	ESBL (58.3)	ESBL, cephalosporin resistance, variable sensitivity to carbapenems	[34,72,85,125]
	Resistance to C3G frequent		
*Enterococcus* spp.	Vancomycin resistance (67.5)	VRE, resistance to glycopeptides, aminopenicillins	[60,67,92]
	Ampicillin resistance frequent		
*Candida* spp.	Fluconazole resistance: variable	Resistance to azoles, sensitivity to amphotericin B	[81,97,122]
	*C. krusei*: intrinsic resistance		
*Salmonella* spp.	ESBL: detected	Resistance to beta–lactams, aminoglycosides, cotrimoxazole	[40,120]
	Multi–resistance ≥ 6 antibiotics		
*Proteus* spp.	Ampicillin resistance (100)	Resistance to beta–lactams, aminopenicillins, quinolones	
	Multi–resistance frequent		

ESBL: Extended–spectrum beta–lactamase–producing enterobacteria; MRSA: Methicillin–Resistant *Staphylococcus aureus*; MDR: Multidrug–Resistant Bacteria; CRE: Carbapenem–Resistant Enterobacteriaceae; C3GR: Resistance to third–generation cephalosporins; CRAB: Carbapenem–Resistant *Acinetobacter baumannii*; VRE: vancomycin–resistant *enterococcus*.

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
