# Peer review of "Surveillance of Healthcare-Associated Infections in the WHO African Region: Systematic Review of Literature from 2011 to 2024"

_antibiotics, 2025, doi:10.3390/antibiotics14121287_

Round 1

Reviewer 1 Report

Comments and Suggestions for Authors

The authors address a high-impact healthcare problem in a region where information is scarce. The long period analysed makes the work even more valuable. A large number of studies were included in the systematic review, and the manuscript provides a considerable body of data. I acknowledge the substantial effort the authors made to retrieve and compile this valuable information.

Nevertheless, the presentation of results is not well organised. In places it is redundant, and in others contradictory. While the discussion appears coherent, the difficulty in interpreting the results as presented prevents the reader from concluding with certainty that the discussion and conclusions accurately reflect the analysis of the 103 studies.

Several flaws are identified; some, but not all, are mentioned under “minor comments.” I strongly suggest that the authors carefully consider both these and other reviewers’ comments in order to improve the presentation of the manuscript so that this valuable body of information can better achieve the study’s stated objectives.

Major comments

  • Definition of HAI: While some literature refers to infections acquired by healthcare staff as HAIs, this is not the standard definition. The authors cite an outdated reference to support this usage. HAIs are defined as infections acquired in healthcare settings by hospitalized patients, usually with underlying conditions and exposed to predisposing factors (catheters, surgery, etc.). This context is very different from that of healthcare workers—who are often otherwise healthy—acquiring occupational infections, frequently associated with unsafe practices such as needle punctures. The physiopathology and epidemiology of these two types of infections differ substantially, and it is uncommon to address HAIs and occupational infections together. The authors should align with current definitions of HAIs as provided by WHO, CDC, ECDC, or other recognized authorities.
  • Introduction:
    • The section is too long and at times redundant. It could be condensed for clarity and to avoid contradictions.
    • The final two paragraphs fit better within the “Discussion/Conclusions” and “Materials and Methods” sections, respectively.
    • The authors state that “in recent years” HAIs have been recognized as a public health problem in developing countries. Maybe they mean that, while the problem has been known for many years, only recently have specific actions (such as surveillance) been implemented. Yet, the cited references (1995, 2005, 2006) do not support the claim of “recent”.
    • The eighth paragraph states that surveillance systems exist in “some developed countries,” while in fact such systems exist in most developed countries and many developing countries. The reference provided (2009) is outdated, and the statement is repeated again later (eleventh paragraph) with a slightly newer but still old reference (2018).
    • It is remarkable that the description of infection types is not included as an objective of the review.
  • Materials and Methods:
    • The main PRISMA reporting guidelines were updated in 2020 and are available at the PRISMA website. The authors cite a 2015 reference instead.
    • The research strategy is confusing. A clearer workflow diagram or narrative would help. Furthermore, the authors claim to have conducted the search “without language restrictions” but later state that only English and French articles were included.
  • Results:
    • Some results are contradictory, e.g., the number of countries represented (South Africa 17 vs. 14; Nigeria 19 vs. 18) and the number of studies using genotypic methods (3 vs. 16).
    • Table 6 totals 68 studies—what about the remaining 35?
    • Antimicrobial resistance is mentioned only briefly in the results, and not clearly. Yet in the discussion, the authors cite precise AMR mechanisms and state that Acinetobacter baumannii and Pseudomonas aeruginosa are multidrug-resistant. While this is frequently true, it is not universal.
  • Discussion:
    • The statement “During our literature review, we identified several studies reporting HAIs linked to epidemics” is not supported by the results, as those studies are not described there.

Minor comments

  • In the first line of the introduction, the acronym “HCAI” is given for healthcare-associated infections, while in the third paragraph the authors state that, “for ease of reading, the term HAI will be used throughout this article”—and indeed they do. To avoid confusion, revise the first line accordingly. Also, once “HAI” is chosen, avoid introducing “nosocomial infection (NI)” (ninth paragraph).
  • Section 2.1: define the acronym “PICO” and provide a reference.
  • Section 2.3, third paragraph: point 2) is unclear to this reviewer and requires clarification.
  • Some methods described (including in tables) are not consistent with the results. For instance, Table 3 does not list “staff” as a potential source of HAIs, yet in Table 6 staff is identified as the most frequent source.
  • In the discussion, when referring to the main agents of HAIs, use the term “species” rather than “strain.”

Reviewer 2 Report

Comments and Suggestions for Authors

In this review manuscript, the authors systematically examine healthcare-associated infection (HAI) surveillance studies conducted in the WHO African Region during 2011- 2024. They find that most investigations were local and cross-sectional, with surveillance largely relying on phenotypic methods and dominated by reports from Nigeria, South Africa, and Ghana. The review underscores the need for standardized approaches to strengthen HAI monitoring and control across the region.

Reviewer’s comments:

Abstract

  1. The “Background” and “Objective” sections partially repeat each other. Consider condensing into one “Background/Objective” so that the objectives are clearly distinguished.
  2. The statement “Despite some progress” is vague: please specify what progress was observed.
  3. While parts of the results (local and cross-sectional study design, limited use of molecular methods, concentration in a few countries) are well reflected in the conclusions, other findings, particularly the distribution of pathogens and sources of HAIs, are not connected to the concluding statements. To strengthen the abstract, I suggest either (a) explicitly linking these epidemiological results to broader implications (e.g., how pathogen distribution or infection sources highlight specific gaps in infection prevention and control) or (b) streamlining the results to focus only on those findings that directly support the conclusions.

Introduction

  1. Redundancy and inconsistent terminology. The term “nosocomial infections (NI)” is defined more than once, and both “NI” and “HCAI” are introduced before later switching to “HAI.” (the 1st and 2nd paragraphs in the Introduction) To improve clarity and readability, I recommend streamlining these definitions and consistently using a single acronym (HAI) throughout the manuscript.
  2. The statement “only a few African countries have established national HAI surveillance systems” is repeated almost word-for-word in the Introduction (8th and 11th paragraphs).

Materials and methods

  1. Exclusion criterion (5) states “articles in languages other than French and English,” but the abstract mentioned “without language restrictions.” This could be confusing. I suggest moving the “without language restrictions.” In the abstract.
  2. Section 2.5 is titled “Assessment of risk of bHAI” – what does this mean? Typos?

Results

  1. Figure 1, is the “eligibility criteria” the same as the five “inclusion criteria”? If so, please keep the terminology consistent.
  2. Table 6, please change comma (,) to dot (.) to show percentages in the last column. Please also check other tables for this issue (e.g., Table 7).
  3. In Table 7, the numbers in the 2nd column sum to 113, exceeding the 103 studies included. This likely reflects overlap, where individual studies reported multiple pathogens. If this is the case, please clarify in the text, or note it right under the table.

Discussions

  1. Similar to what was commented about the Abstract, some descriptive results (e.g., the distribution of pathogens and infection sources) are not clearly tied to the conclusions in the abstract. The discussion should explicitly link these findings to implications for infection control and surveillance priorities.

Reviewer 3 Report

Comments and Suggestions for Authors

The manuscript consists of a systematic review of the literature on HAIs in Africa and the presence and systematic nature of an epidemiological surveillance system. While it incidentally addresses the issue of antibiotic resistance, the topic is more suited to a journal focused on infections or public health in general, though not exclusively.

introduction: revise for consistency and uniformity along the text the use of terminology (HCAI, HAI, NI). Authors mentioned “the term HAI will be used throughout this article”

Methods: The review is robust in its strategy and comprehensive. However, the time range for study inclusion should be clarified, as well as the scope of the databases used (multiple databases are cited with different methodologies). Revise also the title if the range for inclusion is wider (african scientific literature database analysed without time restrictions?)

The bias assessment is mentioned but not included in the file.

PICO repeated 

Results: Clearly reported and well discussed. It would probably be advisable to add a paragraph in the discussion about sources of infection, especially in light of the recommendations. 

Sections 4.5 and 4.6 are clear. “healthcare personnel” 22.3%: how was determined the role of the personnel? exclusion of other sources or hand hygiene or others?. For the hospital environment, was there confirmed contamination in surfaces? Overall, define better in the method these sources as not consistent between methods and results.

References are adequate.

sincerely

Round 2

Reviewer 1 Report

Comments and Suggestions for Authors

Despite the authors’ efforts, it seems that some major comments have not yet been fully addressed. I apologise if my previous remarks were not sufficiently clear.
For instance:

  1. The authors state that “The definition of IAS has been revised and articles dealing with diseases contracted by healthcare professionals in the course of their work have been removed from the list of included articles.” Nevertheless, the definition remains unchanged; in fact, the entire paragraph—including the references—appears identical to that in the initial version.
  2. The expression “in recent years” has been replaced by “relatively recently”, and the references have been modified, but not by those mentioned in the authors’ responses to the reviewers. In fact, the current references are from 2015 (9) and 2017 (10), not from 2022 and 2023 as indicated. Therefore, the references still seem somewhat outdated.
  3. In response to the reviewer’s comment “It is remarkable that the description of infection types is not included as an objective of the review”, the authors indicate that “the objectives and questions have been expanded.” While the objectives have indeed been modified, in the results section infections are still classified only by source (exogenous and endogenous), and there is no description of the origin, site, or type of infection.
  4. The inclusion of antimicrobial resistance mechanisms is a valuable improvement; however, Table 9 combines phenotypic and genotypic profiles, which may not be conceptually consistent.

In addition to these points, I still have the impression that the substantial amount of data gathered by the authors could be presented in a more structured and coherent way. The authors are encouraged to reconsider the overall organisation of the material, so that it more directly responds to the objectives stated in the abstract: the organisation of surveillance, the pathogens involved, and the frequency of multidrug-resistant species.

It might also strengthen the work to consider dividing the manuscript into two or more articles, each focusing on a more specific set of objectives. This could help improve both clarity and analytical depth.

Author Response

Please find the responses to the reviewers' comments in the attached file.

Round 3

Reviewer 1 Report

Comments and Suggestions for Authors

This reviewer acknowledges once again the considerable effort the authors have made to contribute to the body of knowledge on such an important subject. Neverthesless, some flaws are still present. They may not seem important, but I firmly bealive that, for the sake of clarity and precision --wich that should be mainained in scientific literature -- and also for the reputation of both the authors and the journal, these aspects should be carefully revised.

Some of them are indicated here, but throughout the text there are other inconsistencies that I strongly encourage you to revise as well.

It is not worth defining exclusion criteria if they are exactly the opposite of inclusion criteria.

The sentence “Most systems operated on a voluntary basis, monitored the incidence of IAAS….” is not clear to me. Furthermore, it includes an abbreviation not described in the abbreviation’s list, nor previously in the text (IASS) and it probably stands for HAI or NI. Thus, the authors use indistinctly HAI, NI and IASS. Later in the text, and within the same paragraph, IAS and ISA also apperar, wich adds to the confunsion.

When the authors say “This paradgim shift...”, which paradigm are they referring to?

Multi-drug resistant bacteria abbreviated as MRB in the text, while in the abbreviations list the more standard form, MDR, is used.

The sentence “HAIs are underreported” -- is a presumption or is it based on exiting reports or publications? If so, please provide references.

Author Response

Reviewer’s comment /Commentaire du réviseur

Author’s response/ Réponse de l'auteur

Revised text or action taken / Texte révisé ou mesure prise

1

It is not worth defining exclusion criteria if they are exactly the opposite of inclusion criteria.

Only exclusion criteria 1 and 2 mirror the inclusion criteria.

Exclusion criteria 1 and 2 will be removed.

2

The sentence “Most systems operated on a voluntary basis, monitored the incidence of IAAS….” is not clear to me.

This paragraph has been reworded and replaced.

The new wording is as follows: « Most systems operated on a voluntary basis by health services. These systems monitored the incidence of HAIs using the definitions of the Centre for Disease Control and Prevention »

3

Furthermore, it includes an abbreviation not described in the abbreviation’s list, nor previously in the text (IASS) and it probably stands for HAI or NI. Thus, the authors use indistinctly HAI, NI and IASS. Later in the text, and within the same paragraph, IAS and ISA also apperar, wich adds to the confunsion.

We take this comment into consideration.

All IAS and ISA abbreviations have been corrected in the text. Other abbreviations have been updated in the list of abbreviations.

4

When the authors say “This paradgim shift...”, which paradigm are they referring to?

The SENIC project was a starting point for improving the organisation of HAI surveillance in hospitals in the US. Based on these results, HAI surveillance models have been implemented in countries, especially those with limited resources, to optimise patient safety (reference 11).

5

Multi-drug resistant bacteria abbreviated as MRB in the text, while in the abbreviations list the more standard form, MDR, is used.

We take this comment into consideration.

The abbreviation MRB has been replaced by MDR.

6

The sentence “HAIs are underreported” -- is a presumption or is it based on exiting reports or publications ? If so, please provide references.

This is a presumption given the many challenges facing most countries : limited resources; fragile health information systems; competing public health priorities; lack of binding legislation or policies; fear of bad publicity. There are therefore no specific references.